# The Architecture of Built Pedagogy for Active Learning—A Case Study of a University Campus in Hong Kong

**Edmond W.M. Lam** [1], **Daniel W.M. Chan** [2]  **and Irene Wong** [2],*

1   School of Professional Education and Executive Development (SPEED), The Hong Kong Polytechnic University, Hung Hom, Kowloon, Hong Kong; wmedlam@speed-polyu.edu.hk

2   Department of Building and Real Estate, The Hong Kong Polytechnic University, Hung Hom, Kowloon, Hong Kong; daniel.w.m.chan@polyu.edu.hk

*   Correspondence: irene.w@connect.polyu.hk; Tel.: +852-9028-6972

**Abstract:** Traditional teaching modes are engaged with teachers delivering knowledge to students with minimum feedback. Teaching is conducted in lecture theaters and classrooms, which are sometimes designed with minimum flexibility for university education. However, the rapid development of information and communication technologies has altered the teaching pedagogy from traditionally teacher-centered to more collaborative learning between teachers and students. Learning spaces should be designed to be interactive and collaborative with suitable physical movement and social engagement among teachers and students. This paper aims to examine the relationships between modern technology and pedagogical shift, and to identify and discuss the essential design principles for effective active learning through built pedagogy. A recent renovation project of The Hong Kong Polytechnic University in converting conventional classrooms and lecture theaters to active learning spaces was adopted as a case study to illustrate and validate the design principles and their actual implementation. Feedback and responses from 410 end-user students on the impact of the renovated classrooms and lecture theaters on teaching and learning effectiveness were gleaned through empirical survey questionnaires dispatched face-to-face to students after attending classes in the renovated classrooms and lecture theaters. The results of factor analysis indicated that the 15 variables of key design criteria for active learning spaces were consolidated under six underlying clustered factor groups: (1) Versatility of learning space; (2) interior design of learning environment; (3) modern information technology / audio and video (IT/AV) technologies; (4) interior lighting; (5) comfortable furniture and acoustic design; and (6) interior temperature. The survey findings can serve as good references and useful insights for architects in designing new learning spaces and facilities that assist active and collaborative learning for university students in future.

**Keywords:** built pedagogy; active learning; modern technologies; versatility and flexibility; aesthetic

## 1. Introduction

Hiller wrote in the book *Space is the Machine*: "Architecture implies both a space and an activity." Space is the objective property of the building and given significance by linking it directly to human behavior or intentionality [1], and environments can create a relaxing and sociable setting [2]. In other words, functions define a space, and space design facilitates or limits functions. Built pedagogy is architecture of space design, which can define how one learns, teaches, acts, or responds, reflecting the current pedagogy [3–6]. The roles of architects in designing learning spaces is to design and construct a space/building according to the requirements of all stakeholders and responding to the pedagogical

changes, which can facilitate and enhance learning. Brown and Long [7] refer to faculty staff and students as product experts and architects as learning space development experts.

This paper attempts to investigate the recent development of built pedagogy and studies on how the architectural design of learning spaces can facilitate and enhance learning. The design principles of built pedagogy are identified and illustrated through a recent case study of renovating classrooms and lecture theaters of The Hong Kong Polytechnic University in Hong Kong. The success of the renovation project was analyzed based on the students' feedback and comments on the performance of the renovated learning spaces, which were collected by an empirical questionnaire survey with end-user students. The survey findings confirm and validate the identified design principles of active learning spaces.

## 2. Current Pedagogical Changes

Conventional teaching in universities is based on one-way delivery of knowledge from teachers to students [8]. The rapid development of information and communication technologies, together with the advent of the constructivism learning paradigms, have generated a pedagogical shift from conventional one-way teaching to more active and collaborative learning environments [7,9]. Classroom configuration should facilitate teachers to engage with individual students by moving freely around the classroom [10]. Lam et al. [11] identified such learning activities as group debates, forum discussions, and teamwork, which can best be carried out in small discussion groups in learning spaces with lightweight and movable wheeled chairs, allowing flexible seating configurations. A computer network is an effective means of enabling students to access online learning materials, even with real-time interactive communication that can offer a large variety of teaching approaches for teachers [7,8,12,13]. Teaching spaces are not confined to only formal lecture/classrooms, but extended to the whole campus, including informal learning spaces [8]. Facilities that encourage learners' active participation and collaboration are increasingly important in learning space design [6,8].

## 3. Design Principles of Active Learning Spaces

Active learning emphasizes interaction and collaboration among teachers and students. Space design and provided facilities can facilitate collaborative learning, presentation, and group work, as well as enhance concentration in learning. Architects should design learning spaces that can facilitate active learning [14]. Collaborative learning and teamwork require computer support for various needs of content sharing and exchange of information management [13]. Brown and Long [7] identified three major trends in learning space design: (a) Active and social learning strategies that promote students' active and social engagement in learning; (b) human-centered design focusing on user-orientation; and (c) devices that enrich learning, such as modern technologies [7]. Jamieson et al. [8] proposed seven guiding principles in learning space design: (a) Multi-functionality; (b) flexibility; (c) making use of vertical dimensions, e.g., walls for display; (d) integration with campus functions; (e) maximizing teachers' and students' control on environments and facilities; (f) maximizing alignment of different curriculum activities; and (g) maximizing student access to and use of learning outcomes. Brubaker [15], Chiu [10], Cornell [2], Leggett et al. [16], and Monahan [3] added four major design principles for an active learning space, including: (a) Ample working spaces; (b) ability to facilitate student–teacher interaction; (c) creating a comfortable and safe environment; and (d) motivation for learning. These essential design principles have seven implications: (a) Rooms must be wired for the communication and network access; (b) learning spaces are preferred to be multi-functional and convertible for diverse functions; (c) furniture must be redeployed to facilitate computer use, teacher–student interaction, and collaboration among students; (d) lighting must be ambient and glare free; (e) ambient lighting, good sound insulation, and comfortable temperature are vital to learning space design; (f) ability to control internal environment and facilities can facilitate learning; and (g) a pleasant and enjoyable interior can promote learning. Table 1 summarizes and elaborates on these design principles of built pedagogy.

**Table 1.** Essential design principles of built pedagogy.

| | |
|:---:|:---:|
| **1** | **Modern Technologies** |
| 1.1 | Ease of access to use the provided communication and IT facilities |
| 1.2 | User-friendliness of the provided facilities |
| **2** | **Space Design** |
| 2.1 | Versatility: Learning spaces are designed for various usage |
| 2.2 | Convertibility: Learning spaces can be converted into different sizes |
| 2.3 | Flexibility: Furniture design facilitates group discussion |
| **3** | **Comfort and Safety** |
| 3.1 | Comfortable furniture to enhance learning |
| 3.2 | Satisfactory acoustic provision |
| 3.3 | Ambient lighting |
| 3.4 | Comfortable interior temperature |
| 3.5 | Ability to adjust internal environment |
| **4** | **Esthetic** |
| 4.1 | Enjoyable environment |
| 4.2 | Vibrant interior design |
| 4.3 | Interesting textures, patterns, and finishing to break monotony of learning spaces |

## 4. Overview of Learning Space Renovations in The Hong Kong Polytechnic University Campus

In view of the current development of teaching pedagogies, The Hong Kong Polytechnic University (PolyU) has carried out a series of refurbishment work to conventional classrooms and lecture theaters under the PolyU Strategic Plan 2012–2018 (The Plan) since the 2014 summer. The Plan aims to offer students a new experience to study the subjects with multi-media resources online as a supplement to classroom teaching with higher student engagement. Lighting integrates into suspended ceiling design. The metal ceiling panels create a smoothing lighting effect by partially reflecting and diffusing direct light. Adjustable lighting and temperature controls are installed to facilitate different types of presentations and increase learning comfort. A new style of informal learning environment is also created.

*4.1. Upgrading of Classroom (QR-611)*

The classroom (QR-611) was transformed from a single-function computer laboratory to a general teaching room. New technologies like whiteboards, a touch-sensitive monitor, two high-resolution projectors, and a sidewall monitor screen were installed to facilitate active learning. These provisions allow PowerPoint presentations and elaborating lectures on whiteboards to be performed simultaneously. Mobile swivel chairs were provided, which can be rotated to facilitate group discussion. Flooring was designed with vibrant color strips to improve learning incentive. Figure 1 illustrates the upgrading work of QR-611. Both teachers and students are satisfied with the overall learning space design of QR-611.

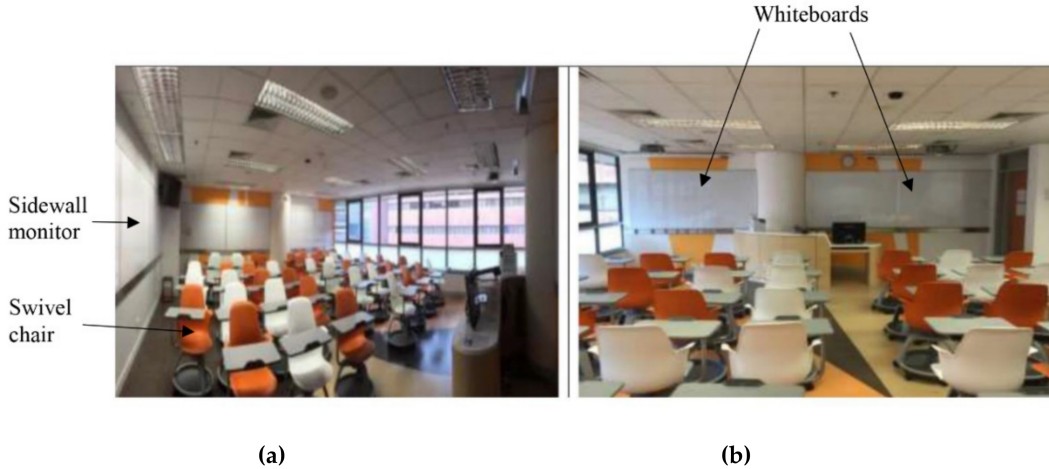

**(a)** **(b)**

**Figure 1.** Upgrading of classroom (QR-611): Original tables and chairs (**b**) replaced by swivel chairs (**a**) to enhance personal comfort and facilitate student grouping.

*4.2. Refitting of Classroom (BC-404)*

The classroom (BC-404) was refitted from a teacher-focused, lecture-style layout to an interactive classroom (Figure 2). The modular tables and mobile chairs facilitate group discussion in different sizes. Lighting is integrated into the ceiling system. The lively floor and ceiling patterns create momentum in learning. Replacing conventional block partitions with glass panels brightens up the learning environment. Both teachers and students welcome the new refreshing space design of the classroom.

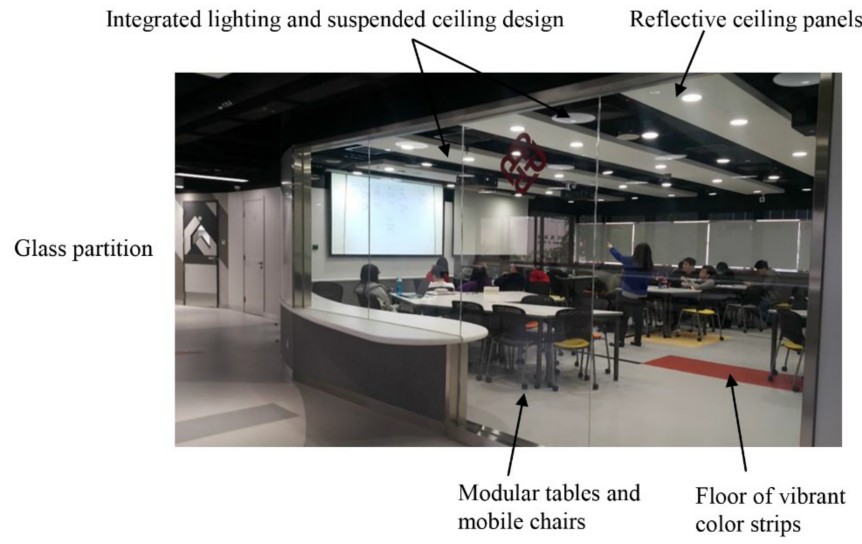

**Figure 2.** Refitting of classroom (BC-404).

*4.3. Reconfiguration of Classrooms (N-001, N-002, and N-003)*

The three classrooms (N-001, N-002, and N-003) were originally a large lecture room. The long-shaped lecture room was subdivided into three smaller classrooms by movable glass partitions, enabling flexible use of space for diverse types of group activities (Figure 3). Modular tables and movable chairs facilitate speedy grouping into different sizes. Installing multiple monitors and writing boards in a single row behind the lectern facilitates teaching in different presentations. The design adds convertibility and flexibility to the original lecture room.

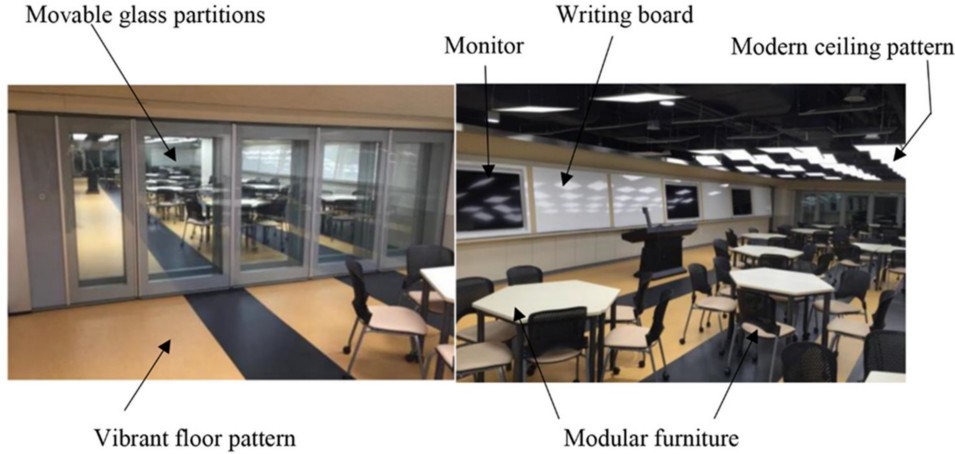

**Figure 3.** Reconfiguring of classrooms (N-001, N-002, and N-003).

*4.4. Refitting of Lecture Theater (TU-201)*

The lecture theater (TU-201) was transformed from a conventional learning space into a modern, comfortable, and technology-enhanced lecture theater for large classes (Figure 4). Advanced information technology / audio and video (IT/AV) facilities, coupled with multiple large screens were installed to facilitate cross-reference presentation, particularly in conducting conferences and seminars. The double-layer furniture setting (one row of tables in two rows of chairs) enable group discussion, which is usually not feasible in conventional lecture theaters. Glass writing panels were installed to sidewalls, allowing the audience to make presentations when necessary and improving interaction between the teacher and students. Both teachers and students are satisfied with the change and concur with the upgrading of facilities for facilitating interactive and collaborative learning, as well as enhancing versatility to the lecture theater.

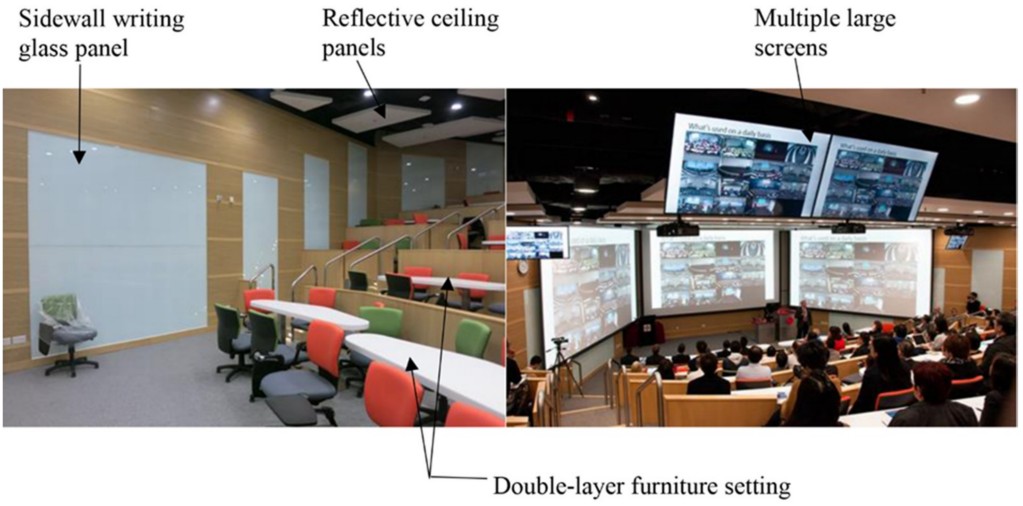

**Figure 4.** Refitting of classroom (TU-201).

## 5. Empirical Questionnaire Survey for End-User Students

Feedback and comments from end-user students on the impact of the renovated classrooms and lecture theaters on teaching and learning effectiveness were solicited via an empirical questionnaire survey for the purpose of assessing the usefulness of the renovation projects at PolyU. Five hundred (500) hardcopies of blank questionnaires were distributed face-to-face to students attending classes in

the renovated classrooms and lecture theaters in November 2017. Four hundred and ten (410) copies of completed questionnaires were collected, with a high response rate of 82%.

## 5.1. Development of Survey Instrument

The empirical survey questionnaire, containing 15 variables of various key design criteria (variables), was developed based on and expanded from the 13 essential design principles under the four categories of "design principles of built pedagogy," which are deduced from extensive desktop literature review and the scope of the PolyU Strategic Plan 2012–2018 (refer to Table 1 and Appendix A for details). Students were invited to rate their levels of agreement on whether the renovated learning spaces achieved the objectives of the identified 15 key design criteria for active learning as listed out on the survey form (Appendix A), according to a 5-point Likert measurement scale (i.e., 1 = strongly agree; 2 = agree; 3 = no comment; 4 = disagree; and 5 = strongly disagree).

## 5.2. Data Analysis of Survey Responses

The common statistical software program (SPSS) was used as the main tool of analysis of survey responses collected in this paper. Factor analysis was applied to reduce the large amount of collected data from the questionnaire survey into smaller manageable sets of components (underlying clustered factor groups) for easier analysis and discussion [17–20].

Principal components analysis for factor extraction, together with the Varimax method of rotation and Kaiser normalization in SPSS FACTOR program were used. Table 2 portrays the communality results after factor extraction, with all the values (0.551–0.932) being larger than the allowable value of 0.2, and thus they all should be included for conducting factor analysis. Table 3 indicates the KMO value of 0.962, equivalent to an "excellent" degree of common variance, which is remarkably higher than the allowable threshold of 0.50 [20,21]—and a very low associated significance level ($p$-value) of 0.000, which validates the application of factor analysis as the analytical tool for such a set of data with sufficient evidence to proceed with further statistical analysis. Variables of similar values were selected and grouped together.

Altogether, six underlying clustered factor groups or components were generated after factor extraction and rotation, as listed in Table 4. The total percentage of variance explained (79.435%) is found to be very good, being considerably greater than the bare minimum of 60% [22,23]. All factor loadings of the 15 individual factors are much higher than 0.50, and significantly above the suppressing value of 0.30 as recommended by Holt [24]. It was manifested that the factor loadings of, and the explanations on, the individual factors extracted are generally congruent and adequate. So, the 15 individual factors (variables) of key design criteria for active learning spaces are wholly included in one of these six underlying clustered factor groups. With the purpose of providing succinct illustrations of the results derived from factor analysis, it was recommended to put forward a symbolic and consolidated name to each of the clustered factor groups with higher values of factor loadings [20,21].

**Table 2.** Communality results of the key design criteria for active learning spaces after factor extraction.

| Variable | Item on Survey Form | Key Design Criteria for Active Learning Spaces | Initial | Extraction |
|---|---|---|---|---|
| V1 | 1.1 | The equipped modern technologies (e.g., computers, projectors, smartboards, video conferencing, 3D visualization, etc.) can assist active learning. | 1.000 | 0.707 |
| V2 | 1.2 | The provision of plug-n-play (e.g., adequate power sockets, internet connection, audio/video connection, multiple monitors, etc.) for using the equipped technologies is useful. | 1.000 | 0.815 |
| V3 | 1.3 | The provided facilities are helpful in modifying, recording, and presenting the retrieved information. | 1.000 | 0.821 |
| V4 | 2.1 | The learning space is designed for various usage (e.g., massive lecture vs. small group discussion for seminar). | 1.000 | 0.785 |
| V5 | 2.2 | The space design facilitates group discussion. | 1.000 | 0.815 |
| V6 | 2.3 | The furniture can be easily reconfigured to facilitate grouping in different sizes. | 1.000 | 0.809 |
| V7 | 3.1 | Moveable chairs equipped with flexible back and adjustable seat height can improve learning comfort. | 1.000 | 0.866 |
| V8 | 3.2 | The acoustic room design is satisfactory. | 1.000 | 0.765 |
| V9 | 3.3 | Lighting is ambient. | 1.000 | 0.789 |
| V10 | 3.4 | Adjustable lighting level can enhance learning comfort. | 1.000 | 0.856 |
| V11 | 3.5 | The interior temperature is comfortable. | 1.000 | 0.932 |
| V12 | 3.6 | Adjustable interior temperature can enhance learning comfort. | 1.000 | 0.551 |
| V13 | 4.1 | The interior learning environment is enjoyable. | 1.000 | 0.713 |
| V14 | 4.2 | Vibrant colors can motivate learning. | 1.000 | 0.838 |
| V15 | 4.3 | The textures, patterns, and finishing are appealing, which can motivate learning. | 1.000 | 0.854 |

**Table 3.** Results of the KMO Test and Bartlett's Test of Sphericity.

| | | |
|---|---|---|
| Kaiser-Meyer-Olkin (KMO) Measure of Sampling Adequacy | | 0.962 |
| Bartlett's Test of Sphericity | Approximate Chi-Square Value | 3648.743 |
| - | Degree of Freedom (df) | 120 |
| - | Significance Level (*p*-value) | 0.000 |

**Table 4.** Results of factor analysis on the 15 variables of key design criteria for active learning spaces.

| Variable | Item on Survey Form | Key Design Criteria for Active Learning Spaces | Factor Loading | Percentage of Variance Explained | Cumulative Percentage of Variance Explained |
|---|---|---|---|---|---|
| **Clustered Factor Group 1 (Component 1): Versatility of Learning Space** | | | | | |
| V6 | 2.3 | The furniture can be easily reconfigured to facilitate grouping in different sizes. | 0.808 | 52.273 | 52.273 |
| V5 | 2.2 | The space design facilitates group discussion. | 0.792 | - | - |
| V4 | 2.1 | The learning space is designed for various usage (e.g., massive lecture vs. small group discussion for seminar). | 0.757 | - | - |

**Table 4.** *Cont.*

| Variable | Item on Survey Form | Key Design Criteria for Active Learning Spaces | Factor Loading | Percentage of Variance Explained | Cumulative Percentage of Variance Explained |
|---|---|---|---|---|---|
| **Clustered Factor Group 2 (Component 2): Interior Design of Learning Environment** | | | | | |
| V15 | 4.3 | The textures, patterns, and finishing are appealing, which can motivate learning. | 0.836 | 7.760 | 60.032 |
| V14 | 4.2 | Vibrant colors can motivate learning. | 0.835 | - | - |
| V13 | 4.1 | The interior learning environment is enjoyable. | 0.548 | - | - |
| V12 | 3.6 | Adjustable interior temperature can enhance learning comfort. | 0.525 | - | - |
| **Clustered Factor Group 3 (Component 3): Modern IT/AV Technologies** | | | | | |
| V2 | 1.2 | The provision of plug-n-play (e.g., adequate power sockets, internet connection, audio/video connection, multiple monitors, etc.) for using the equipped technologies is useful. | 0.787 | 5.895 | 65.927 |
| V3 | 1.3 | The provided facilities are helpful in modifying, recording, and presenting the retrieved information. | 0.764 | - | - |
| V1 | 1.1 | The equipped modern technologies (e.g., computers, projectors, smartboards, video conferencing, 3D visualization, etc.) can assist active learning. | 0.670 | - | - |
| **Clustered Factor Group 4 (Component 4): Interior Lighting** | | | | | |
| V10 | 3.4 | Adjustable lighting level can enhance learning comfort. | 0.789 | 5.170 | 71.097 |
| V9 | 3.3 | Lighting is ambient. | 0.730 | - | - |
| **Clustered Factor Group 5 (Component 5): Comfortable Furniture and Acoustic Design** | | | | | |
| V7 | 3.1 | Moveable chairs equipped with flexible back and adjustable seat height can improve learning comfort. | 0.850 | 4.344 | 75.441 |
| V8 | 3.2 | The acoustic room design is satisfactory. | 0.685 | - | - |
| **Clustered Factor Group 6 (Component 6): Interior Temperature** | | | | | |
| V11 | 3.5 | The interior temperature is comfortable. | 0.877 | 3.994 | 79.435 |

## 6. Findings and Discussions

By summing up Section 3 (Design Principles of Active Learning Spaces), the facilitating factors can be classified into six main factor categories, namely: Factor A—Flexibility and Adaptability; Factor B—User-Friendliness; Factor C—Facilitating Student–Teacher Interaction; Factor D—Comfort and Safety; Factor E—Psychological Appeal; and Factor F—Application of Modern Technologies [2,10,25,26]. Classrooms should be designed to bring together technology, contents, and services in a physical setting, which can promote collaborative learning among students; be equipped with a bundle of technologies in assisting computer-based activities; facilitate group work in different sizes; and allow flexibility and support multi-functioning of the learning spaces [27]. A pleasant learning environment can be created by adequate lighting, effective sound insulation, and adjustable room temperature, while students' learning can be motivated via vibrant colors, attractive textures, and interesting patterns of the learning space [28,29].

Table 4 summarizes the overall results of factor analysis on the 15 variables of key design criteria for active learning spaces. The six components generated include: (1) Versatility of Learning Space; (2) Interior Design of Learning Environment; (3) Modern IT/AV Technologies; (4) Interior Lighting; (5) Comfortable Furniture and Acoustic Design; and (6) Interior Temperature. Active learning emphasizes interaction and collaboration among teachers and students. Learning space design and associated facilities can indeed facilitate collaborative learning, presentation, and group work, as well as enhance concentration in learning.

*Component 1: Versatility of Learning Space*

As group activities are a vital element of active learning, space configuration of classrooms should be able to support collaborative learning. The design of classrooms and lecture theaters should be flexible and encourages students' participation in class [8]. Various research studies have demonstrated that modular furniture can enhance reconfiguration for facilitating group discussion [5,10,29]. The use of modular tables and movable chairs, as demonstrated in the PolyU Strategic Plan 2012–2018, can facilitate speedy grouping into various sizes in classrooms and lecture theaters for launching group discussion. A multi-functional learning space is preferred, which can enable different types of learning activities to be conducted (e.g., large class lectures vs. small group tutorials for presentation and discussion). Movable partitions and flexible furniture allow reconfiguration of spaces for different usage. This component is consistent with Factors A and C.

*Component 2: Interior Design of Learning Environment*

The PolyU Strategic Plan 2012–2018 has indicated that a pleasant, comfortable, and appealing environment can indeed motivate students' learning. This component is supported by the researches of Lippincott [28] and Taylor [29] that lively colors, interesting textures, and appealing patterns can motivate learning. Vibrant colors, interesting patterns, and comfortable textures are applied to the interior of learning spaces, creating a pleasant and enjoyable learning environment. Component 2 is in line with Factor E.

*Component 3: Modern IT/AV Technologies*

Classrooms should be designed to bring together technology, contents, and services in a physical setting, which can promote collaborative learning among students [13]. This component indicates that modern technologies play an important role in teaching pedagogy and can enhance active learning of students. Communication and IT facilities (e.g., computers, projectors, smartboards, video conferencing, 3D visualization, etc.) have become standard provisions to classrooms and lecture theaters at universities. Multiple monitors, touch-sensitive monitors, projector screens, writing glass panels, which can all facilitate teaching by teachers and presentation by students, should be user-friendly. Component 3 corroborates Factor F.

*Component 4: Interior Lighting*

The internal environment of learning spaces affects students' concentration in learning. A comfortable environment with ambient lighting contributes to learning comfort. The ability of users to adjust lighting level enhances different modes of learning activities (e.g., large class lectures vs. small group tutorials for presentation and discussion). Component 4 is consistent with Factor D.

*Component 5: Comfortable Furniture and Acoustic Design*

Mobile chairs equipped with flexible backs and adjustable seat height can improve learning comfort and enhance concentration. Tables and chairs should be designed according to human ergonomics. Proper acoustic design of classrooms is also conducive to better concentration with interactive learning environment for students. Component 5 is in line with Factor D.

*Component 6: Interior Temperature*

The feeling of comfort may vary under different climatic conditions. For instance, a brighter and warmer interior temperature is preferred in the cold winter season. The ability of users to adjust the interior temperature of learning spaces according to their needs at different times can ensure a comfortable, pleasant, and optimal environment for students.

With regard to the relationships between modern technology and pedagogical shift, the design of a learning space should facilitate the convenient use of communication and IT facilities. Prominent spaces are reserved for installation of monitors and projectors. Cables are to be laid in concealed conduits without jeopardizing users' safety. A raised floor is a good design to allow conduits and wiring running below floor with the flexibility for future modification. The sizes of classrooms and lecture theaters should be spacious to accommodate students grouping into different sizes. Long and narrow configuration is not preferred. Over-sized learning spaces can be modified into several smaller classroom/activity rooms by movable partitions for multi-functional uses. Modular tables and mobile chairs should be used to facilitate different group activities. Scientific studies have shown that colors affect our mood and productivity. Applying vibrant colors of stimulating hues in the interior design can increase output [22]. The interior should be designed in a lively and interesting style in soft color tones, with patches of vibrant color only as highlights in order not to distract students' attention in learning. Component 6 corroborates Factors B and D.

## 7. Summary and Conclusions

A recent prompt development of information and communication technologies has led to a significant pedagogical change in teaching and learning environment at universities. A radical pedagogy shift from the traditional passive teacher-focused teaching to a more active student-centered learning has emphasized interactive learning environments between teachers and students, and collaborative group efforts among students. Learning spaces should be designed with the ability of minimizing the physical barriers between teachers and students, and facilitating active learning. Four design principles with thirteen applications for built pedagogy to facilitate active learning were identified from a desktop literature review. The four design principles include: (a) Modern technologies; (b) space design; (c) comfort and safety; and (d) esthetic.

An in-depth case study of the PolyU Strategic Plan 2012–2018 has authenticated the four suggested design principles, which can serve as recommended guidelines for architects in designing effective learning spaces in future. The research findings have not only assisted in upgrading and creating interactive learning spaces and innovative learning facilities, but have also driven profound innovations and improvements of our teaching and learning environment at university campuses, both locally and internationally.

The limitation of this study is that is has been confined to some renovated interactive classrooms or lecture theaters located at PolyU based on the feedback and responses from the end-user students only, but not the teachers, whose opinions and comments should be captured as well. More similar studies should be launched to cover other local universities and overseas for comparison and benchmarking purposes. Further studies should also explore the impact of the learning space design and installed facilities of renovated classrooms on the teaching effectiveness of teachers and learning experience of students via a series of empirical surveys or focus group meetings with these users.

The atmosphere of a learning space can be considered as a microclimatic environment and should be designed to minimize its impact on our environment. Further research is recommended to apply the design principles of learning spaces to achieve sustainability. Design strategies, as laid down by the Department of Energy of the United States [23], which are listed below, can be introduced to the architectural design of built pedagogy and as design guidelines for future renovation work of learning space at PolyU.

- Reduce cooling/heating loads by providing insulation to walls.
- Eliminate glare by installing adjustable blinds.
- Use natural lighting as far as possible, such as high windows, that will not distract student attention to external environment.
- Facilitate natural ventilation by ventilation shaft.

**Author Contributions:** Conceptualization, E.W.M.L., I.W., and D.W.M.C.; validation, E.W.M.L., I.W., and D.W.M.C.; formal analysis, E.W.M.L. and I.W.; investigation, I.W.; resources, D.W.M.C.; data curation, I.W.; writing—original draft preparation, I.W.; writing—review and editing, D.W.M.C.; supervision, E.W.M.L.; project administration, D.W.M.C.; funding acquisition, D.W.M.C.

**Funding:** This project was funded by the Working Group on Innovative Learning Spaces of the Hong Kong Polytechnic University, Hong Kong.

**Acknowledgments:** The authors wish to extend their greatest appreciation to the Working Group on Innovative Learning Spaces of the Hong Kong Polytechnic University (PolyU) for offering financial support to facilitate this research study (Project Account Code: 9A5Z). Furthermore, the authors would like to express their heartfelt thanks to all of those participating students in the empirical questionnaire survey. The generous assistance in distributing and collecting the survey forms completed by the student end-users from the two part-time student assistants (Pekky Chan and Marcus Yang) is gratefully acknowledged as well.

**Conflicts of Interest:** The authors declare no conflicts of interest.

## Appendix A —Template of Survey Form

*Project Title: Impact of Learning Space on Teaching and Learning Effectiveness (Student Survey at PolyU)*

Active learning emphasizes interaction and collaboration among teachers and students. Learning space design and associated facilities can facilitate students' collaborative learning, presentation, and group work, as well as enhance concentration in learning, with the support of a variety of modern IT/AV technologies to facilitate effective teaching by teachers and active learning by students.

This survey aims to collect valuable feedback and comments on the impact of learning space design of those "renovated" classrooms or lecture theaters in assisting active learning based on your learning experience as the student end-users. Your comments will help update, upgrade, and create innovative learning spaces and facilities at PolyU, and drive innovations and improvements in our learning and teaching environment.

*Part A—Opinions on Learning Space Design*

Please rate your level of agreement on the following statements by ticking the appropriate boxes. [1 = strongly agree; 2 = agree; 3 = no comment; 4 = disagree; 5 = strongly disagree]

| | Items | 1 | 2 | 3 | 4 | 5 |
|---|---|---|---|---|---|---|
| **1** | **Modern Technologies** | | | | | |
| 1.1 | The equipped modern technologies (e.g., computers, projectors, smartboards, video conferencing, 3D visualization, etc.) can assist active learning. | □ | □ | □ | □ | □ |
| 1.2 | The provision of plug-n-play (e.g., adequate power sockets, internet connection, audio/video connection, multiple monitors, etc.) for using the equipped technologies is useful. | □ | □ | □ | □ | □ |
| 1.3 | The provided facilities are helpful in modifying, recording, and presenting the retrieved information. | □ | □ | □ | □ | □ |

| | Items | 1 | 2 | 3 | 4 | 5 |
|---|---|---|---|---|---|---|
| **2** | **Flexibility of Space Design** | | | | | |
| 2.1 | The learning space is designed for various usage (e.g., massive lecture vs. small group discussion for seminar). | ☐ | ☐ | ☐ | ☐ | ☐ |
| 2.2 | The space design facilitates group discussion. | ☐ | ☐ | ☐ | ☐ | ☐ |
| 2.3 | The furniture can be easily reconfigured to facilitate grouping in different sizes. | ☐ | ☐ | ☐ | ☐ | ☐ |
| **3** | **Comfort** | | | | | |
| 3.1 | Moveable chairs equipped with flexible back and adjustable seat height can improve learning comfort. | ☐ | ☐ | ☐ | ☐ | ☐ |
| 3.2 | The acoustic room design is satisfactory. | ☐ | ☐ | ☐ | ☐ | ☐ |
| 3.3 | Lighting is ambient. | ☐ | ☐ | ☐ | ☐ | ☐ |
| 3.4 | Adjustable lighting level can enhance learning comfort. | ☐ | ☐ | ☐ | ☐ | ☐ |
| 3.5 | The interior temperature is comfortable. | ☐ | ☐ | ☐ | ☐ | ☐ |
| 3.6 | Adjustable interior temperature can enhance learning comfort. | ☐ | ☐ | ☐ | ☐ | ☐ |
| **4** | **Esthetic** | | | | | |
| 4.1 | The interior learning environment is enjoyable. | ☐ | ☐ | ☐ | ☐ | ☐ |
| 4.2 | Vibrant colors can motivate learning. | ☐ | ☐ | ☐ | ☐ | ☐ |
| 4.3 | The textures, patterns, and finishing are appealing, which can motivate learning. | ☐ | ☐ | ☐ | ☐ | ☐ |

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
