# Peer review of "The Architecture of Built Pedagogy for Active Learning—A Case Study of a University Campus in Hong Kong"

_buildings, doi:10.3390/buildings9110230_

Round 1
Reviewer 1 Report
The research examined the design principles affecting active learning in classrooms of the Hong Kong Polytechnic University. Followings are the suggestions to improve the quality of the research:
1) The in-text citations in the whole manuscript should be revised. For example,
Page 1 line 31, change [3] [4] [5] [6] to [3-6]
Page 1 line 33, change Brown and Long to Brown and Long [7]
2) The communality results of the factor analysis should be shown. Those less than 0.2 should be removed.
3) The total variance explained was not mentioned to explain the percentage of the variance of 6 components.
4) The suppressing factor loading was also not mentioned. The value is recommended as 0.3. If the suppressing factor loading was used, factors 1.1, 1.2, 1.3, 2.1, 2.2, 2.3, 3.3, 3.4 would be removed. In other words, those factors do not have impacts on active learning.
5) Why "no" in Table 3 has the factor loading of 0.994?
6) Why factor 1.1 was written as 1.000....1 in Table 3? same for 2.2, 2.3 and 4.1?
7) The results in Table 4 is inaccurate. For example, factors 3.6, 4.1, 4.2, 4.3 should be in component 1, based on the results of the factor analysis. Factor 1.2 has similar factor loadings in component 1 and 3, why it was selected for component 1?
8) The discussion of the research should be strengthened. There is no comparison or using the research results from previous research to support this research.
9) The limitations of the research is also missing.
Author Response
Please see the attachment, thanks.

Reviewer 2 Report
General comments:
Paper is interesting and it addresses the up to date problem on an example of real building object renovation.
The photographs of the previous design are missing while they are obviously expected when upgrading of the design is discussed.
Detailed comments:
1) line 15: replace „purpots” with ”aims” or “attempts”
2) In section 1 add the short explanation/definition of the term “built pedagogy”
3) line 57-60: This statement is unclear. Some data are not discussed at the reviewing stage, so they cannot be assessed? Which data?
4) line 89: change “false ceiling” into “suspended ceiling”
5) line 99: change “nodal chairs” into “swivel chairs”
6) Fig. 2: change “false ceiling” into “suspended ceiling”
7) line 146: Annex 1 is missing (Annex A?)
There are 13 design principles in the Table 1, while in the Legend (line 170) there are 15. Needs either unification or explanation.
8) line 170: change “facilities is” into “facilities are”
9) Legend: wrong numbering – doubled 4.1
10) line 171 Section 6 – finding and discussion fails to present the content with reference to the research aim of this article.
11) line 175 – In the Table 4 there are headers; the relationship between them is insufficiently explained
12) line 208: section 3. Conclusion- wrong numbering; it should be number 7.
The title of this section is inadequate to the content. “Summary” would be better title in this case.
